# Descriptive transcriptome analysis of tendon derived fibroblasts following *in-vitro* exposure to advanced glycation end products

Shivam H. Patel[1], Christopher L. Mendias[2,3], Chad C. Carroll[1] *

1 Department of Health and Kinesiology, Purdue University, West Lafayette, IN, United States of America,
2 Hospital for Special Surgery, New York, NY, United States of America, 3 Department of Physiology and Biophysics, Weill Cornell Medical College, New York, NY, United States of America

* carrol71@purdue.edu

## Abstract

### Background

Tendon pathologies affect a large portion of people with diabetes. This high rate of tendon pain, injury, and disease appears to manifest independent of well-controlled HbA1c and fasting blood glucose. Advanced glycation end products (AGEs) are elevated in the serum of those with diabetes. *In vitro*, AGEs severely impact tendon fibroblast proliferation and mitochondrial function. However, the extent that AGEs impact the tendon cell transcriptome has not been evaluated.

### Objective

The purpose of this study was to investigate transcriptome-wide changes that occur to tendon-derived fibroblasts following treatment with AGEs. We propose to complete a descriptive approach to pathway profiling to broaden our mechanistic understanding of cell signaling events that may contribute to the development of tendon pathology.

### Methods

Rat Achilles tendon fibroblasts were treated with glycolaldehyde-derived AGEs (200μg/ml) for 48 hours in normal glucose (5.5mM) conditions. In addition, total RNA was isolated, and the PolyA$^+$ library was sequenced.

### Results

We demonstrate that tendon fibroblasts treated with 200μg/ml of AGEs differentially express 2,159 gene targets compared to fibroblasts treated with an equal amount of BSA-Control. Additionally, we report in a descriptive and ranked fashion 21 implicated cell-signaling pathways.

### Conclusion

Our findings suggest that AGEs disrupt the tendon fibroblast transcriptome on a large scale and that these pathways may contribute to the development and progression of diabetic

**Data Availability Statement:** All relevant data are available from the GEO database at accession number GSE204714.

**Funding:** This work was supported by NIH pre-doctoral fellowship F31-AR073647 (S.H.P), Purdue University Research Initiative Funds (C.C.C), and Ralph W. and Grace M. Showalter Research Trust Award (C.C.C.). The sponsors did not have a role in the study design, data collection and analysis, decision to publish, or preparation of the manuscript.

**Competing interests:** The authors have declared that no competing interests exist.

tendinopathy. Specifically, pathways related to cell cycle progression and extracellular matrix remodeling were affected in our data set and may play a contributing role in the development of diabetic tendon complications.

## Introduction

Tendon degeneration and impaired biomechanical function result in significant reductions in mobility and quality of life for the majority of the ~30 million Americans living with diabetes, resulting in a substantial economic burden to individuals and society. Compounding the problem, human [1–3] and rodent [4] studies indicate that improving blood glucose levels does not normalize tendon properties in individuals with diabetes. Any new approach to enhance tendon health in people with diabetes is hindered by a poor understanding of the underlying etiology of tendon degeneration and impaired biomechanical properties [1, 2, 5–7].

Our previous cell culture work implicated advanced glycation end-products (AGEs) as a potential mechanism driving tendon degeneration [8]. AGEs can form non-enzymatic cross-links with collagen [9], a mechanism that has traditionally been the focus of tendon complications in persons with diabetes [10]. Yet, recent studies of tendons from humans with diabetes have found no evidence of greater collagen crosslinking than those without diabetes [1, 11] and no relationship between tendon AGE content and tensile mechanics [11]. A less explored mechanism of AGE-mediated effects is the interaction of serum AGEs with AGE receptors (RAGE). AGEs accumulate in the serum of patients with diabetes [12–14] and our cell culture data suggest that AGEs can impact tendon cells. Specifically, treatment of cells with AGEs dose-dependently reduced cell proliferation and mitochondrial ATP production.

A thorough understanding of the cell signaling events contributing to the development of AGE-mediated diabetic tendinopathies will assist in exploring alternative areas of thought and developing therapeutic options to target this large patient population. Therefore, to better understand the effect of AGEs on tendon cells, we sought to characterize the alterations to the tendon fibroblast transcriptome following exposure to AGEs. Although many of these pathways have already been implicated with AGEs from analysis of non-tendon tissues, the primary goal of this study was to establish a descriptive and ranked evaluation of pathway disruptions that occur to tendon fibroblasts following an AGE insult.

## Materials and methods

### Animal protocol

Animals utilized in this study were from a previous investigation [8]. The study was approved by the Purdue University Institutional Animal Care and Use Committee. All animals were cared for per the recommendations in the Guide for the Care and Use of Laboratory Animals. Five eight-week-old female Sprague-Dawley rats were purchased from Charles River Laboratories (Wilmington, MA) and maintained for an additional eight weeks. Rats were housed on a 12-hour light-dark cycle and provided standard rat chow and water ad libitum. At sixteen weeks (Final Weights: 256.43±5.19 g), rats were euthanized by decapitation after $CO_2$ inhalation.

### Tendon fibroblast isolation and cell culture

Tendon-derived fibroblasts utilized in this study were from a previous investigation [8]. Briefly, Achilles tendons were rinsed with sterile PBS, minced, placed in DMEM containing

0.2% type I collagenase, and incubated in a 37˚C shaking water bath for four hours. After digestion, the cell suspension was filtered through a 100μm mesh filter, pelleted by centrifugation, and resuspended in 5.5mM glucose DMEM containing 10% FBS, 1% sodium pyruvate (Sigma, St. Louis, MO), and 1% penicillin/streptomycin (Thermo Scientific, Waltham, MA). Samples were then plated in 100mm collagen-coated dishes. After reaching confluency, tendon fibroblasts were split and seeded (100,000 cells) in 100mm collagen-coated culture plates. Each donor animal's (n = 5) tendon fibroblasts were treated separately with 200μg/ml of BSA-Control or AGE-BSA for 48 hours for downstream paired DESeq2 analysis. Tendon fibroblasts treated at passages 2–4 were used for RNA isolation and RNA-sequencing (RNAseq).

## Age preparation

Details on the preparation of AGEs have been reported previously [8, 15]. Briefly, sterile filtered 30% BSA solution (Sigma, St. Louis, MO) was incubated with 70mM glycolaldehyde dimer (Sigma) in sterile PBS for three days at 37˚C. After incubation, the AGE product was dialyzed against sterile PBS for 24 hours at 4˚C using gamma-irradiated 10kDa cut-off cassettes (Thermo Scientific, Waltham, MA) to remove unreacted glycolaldehyde. Unmodified control BSA was prepared similarly, without the addition of glycolaldehyde dimer. Protein concentration was determined by BCA assay (Thermo Scientific) and absence of endotoxin (<0.25Eu/ml) was confirmed via the LAL gel-clot assay (GenScript, Piscataway, NJ).

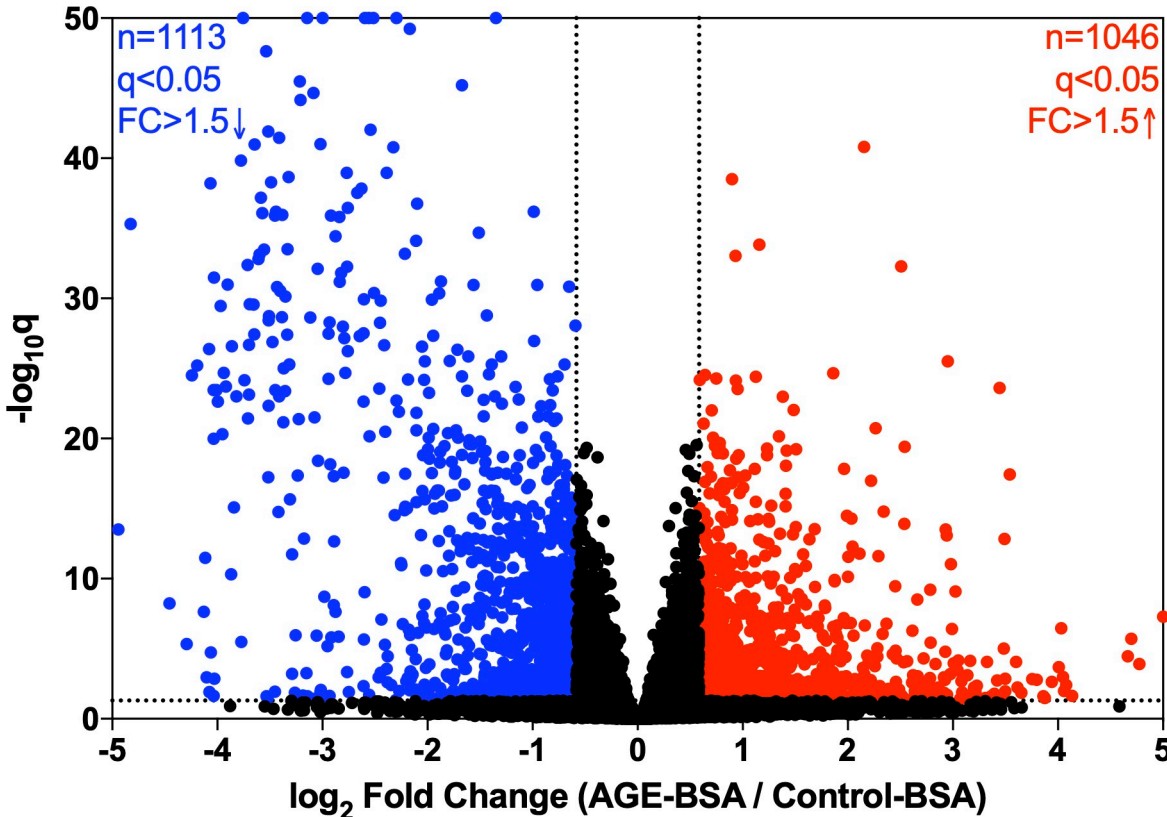

**Fig 1. Volcano plot overview of RNA sequencing results.** Each point represents a single gene target. Red (n = 1046) indicates significant increase in gene expression. Blue (n = 1113) indicates significant decrease in gene expression. Black (n = 10,648) indicates gene targets that were either unaltered or did not meet our thresholds of q<0.05 and fold change of greater that 1.5 or less than -1.5.

The extent of BSA modification was confirmed by fluorescence, absorbance, and loss of primary amines [15–18]. AGE-BSA and Control-BSA were diluted to 1mg/ml in PBS and fluorescent spectra and absorbance were recorded at 335nm excitation/420nm emission and 340nm, respectively (Molecular Devices, San Jose, CA). For determination of loss of primary amines AGE-BSA and Control-BSA were diluted to 0.2mg/ml in PBS. An equal volume of ortho-phthalaldehyde solution (Sigma) was added and fluorescent spectrum was recorded at 340nm excitation/455nm emission (Molecular Devices). Absorbance readings were completed to determine the extent of glycation. AGE-BSA showed increased glycation with absorbance readings of 0.682 AU compared to 0.01 AU for control BSA. AGE-BSA primary amine terminals underwent complete modification (-0.03% accessible amine terminals remaining), while control BSA retained 81.48% of accessible amine terminals. Negative values were interpreted as zero, and extent of modification was similar to previous reports [15].

## RNA sequencing

Total RNA was isolated as previously described [8]. Briefly, RNA was isolated after BSA-Control or AGE-BSA treatment using the Direct-zol RNA Miniprep kit (Zymo Research, Irvine, CA). On-column DNase digestion was completed on all samples before elution of RNA. Total RNA from BSA-Control (n = 5) and AGE-BSA (n = 5) treated tendon fibroblasts was submitted to the Purdue University Genomics Core Facility (West Lafayette, IN) for PolyA$^+$ library construction. The integrity of input total RNA was assessed using a Bioanalyzer RNA Nano chip (Agilent 2100, Santa Clara, CA). Libraries from 500ng of input total RNA were constructed as directed by the Nugen Universal Plus mRNA-Seq + UDI kit (PN#9144–96), but the RNA fragmentation time was decreased from 8 minutes to 4 minutes. Final library products were subjected to a 0.7 Ampure:1 Sample ratio purification to reduce lower molecular weight

**Table 1. Most affected gene targets.**

| Gene | log$_2$ Fold Change | q Value |
|---|---|---|
| Cyp1a1 | 7.07 | 6.77E-07 |
| Pipox | 4.78 | 7.59E-04 |
| Btc | 4.70 | 1.87E-05 |
| Slc22a14 | 4.66 | 2.49E-04 |
| Tbxas1 | 4.08 | 1.46E-02 |
| Itgb2 | 4.06 | 3.42E-02 |
| Slc13a3 | 4.05 | 4.78E-03 |
| Cldn1 | 4.03 | 3.80E-06 |
| Ncf1 | 4.01 | 1.19E-03 |
| Tnfrsf17 | 3.94 | 9.65E-03 |
| Pimreg | -4.94 | 1.01E-12 |
| Pmch | -4.83 | 1.49E-33 |
| E2f7 | -4.45 | 8.80E-08 |
| Pbk | -4.29 | 3.99E-05 |
| Parpbp | -4.24 | 3.47E-23 |
| Ube2c | -4.19 | 7.41E-24 |
| Troap | -4.13 | 3.23E-07 |
| Cenpf | -4.12 | 8.15E-11 |
| Cldn23 | -4.10 | 4.98E-03 |
| Ccnb2 | -4.08 | 5.44E-25 |

amplicons. The resulting libraries were assessed with an Agilent DNA High Sensitivity Chip for yield and quality and sequenced by Novogene (Sacramento, CA). Ten libraries were pooled

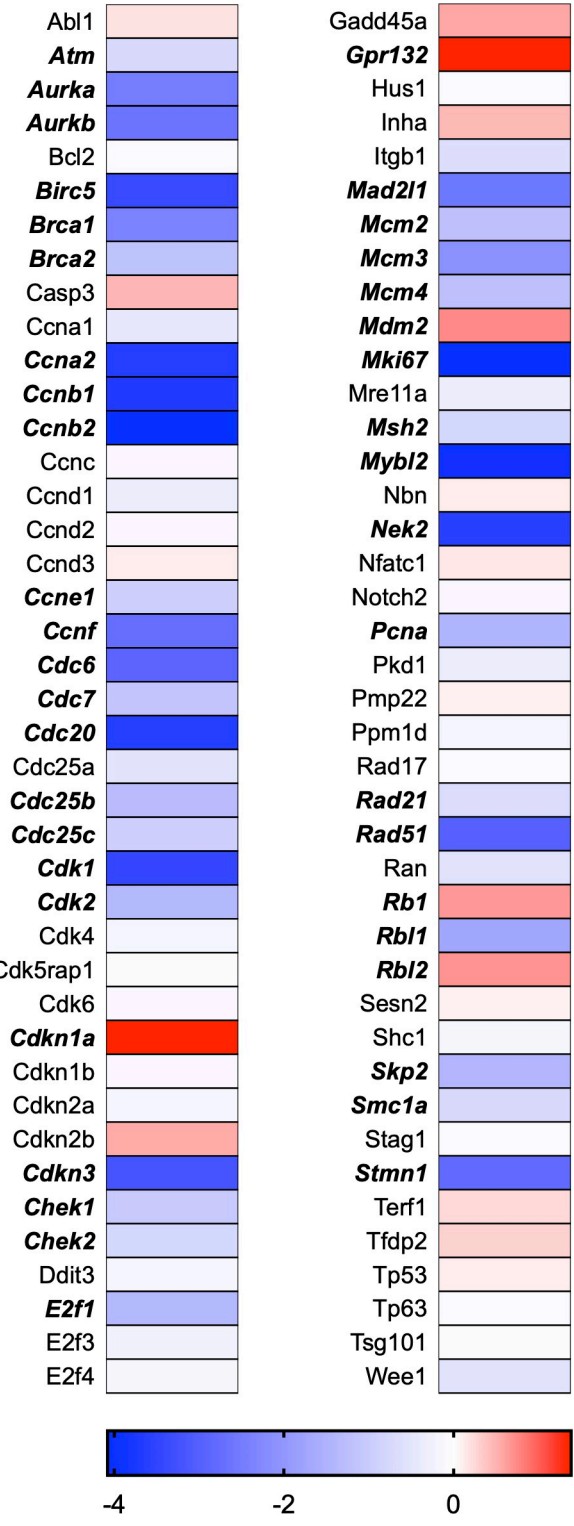

**Fig 2. Cell cycle heat map.** Bold text indicates significantly altered gene targets.

and evenly distributed across a single HiSeq lane to generate ~40,000,000 2X150bp reads on the HiSeq 4000 platform (Illumina, San Diego, CA).

## Bioinformatics

RNAseq raw data set quality and analysis was completed using Basepair software (New York, NY) pipelines. Reads were first aligned to the transcriptome derived from rn6 genome assembly using STAR with default parameters [19]. Next, read counts for each transcript were measured using featureCounts, and differentially expressed genes were determined using DESeq2 using a paired analysis [20, 21]. An adjusted p-value cut-off of 0.05 (corrected for multiple hypotheses testing) was used. Finally, GSEA was performed on normalized gene expression

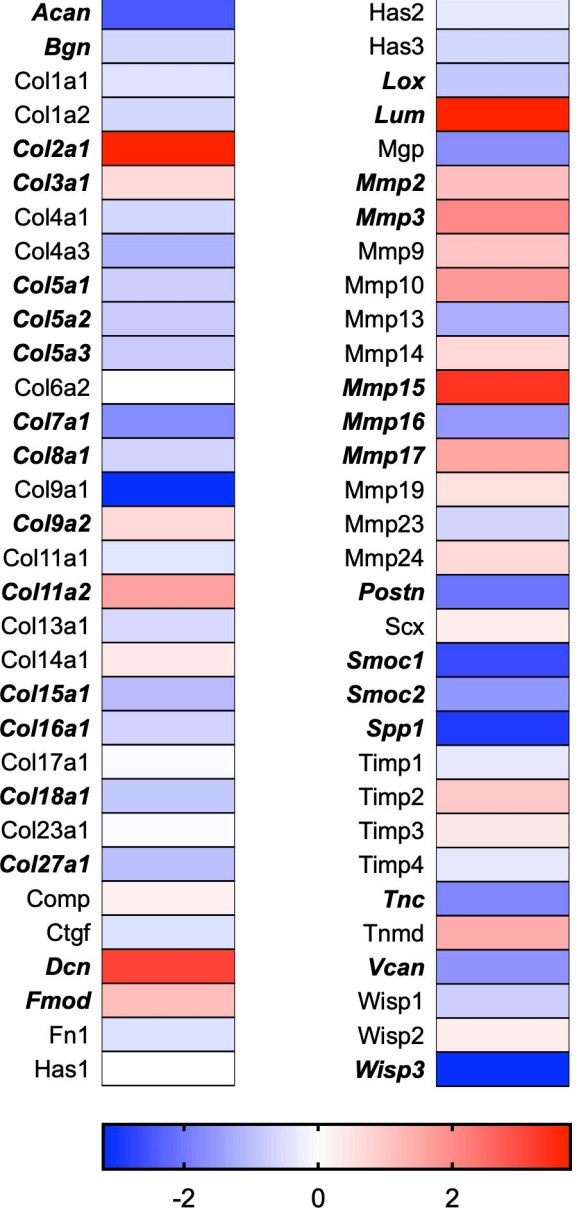

**Fig 3. ECM and tenogenic markers heat map.** Bold text indicates significantly altered gene targets.

counts, using gene permutations for calculating p-value. A log$_2$ fold change cut-off of 1.5 was enforced.

## Descriptive pathway profiling

To preserve unbiased gene target selection and maintain a hypothesis-driven pathway selection, GeneGlobe (Qiagen, Hilden, Germany) pathway database was utilized to complete a

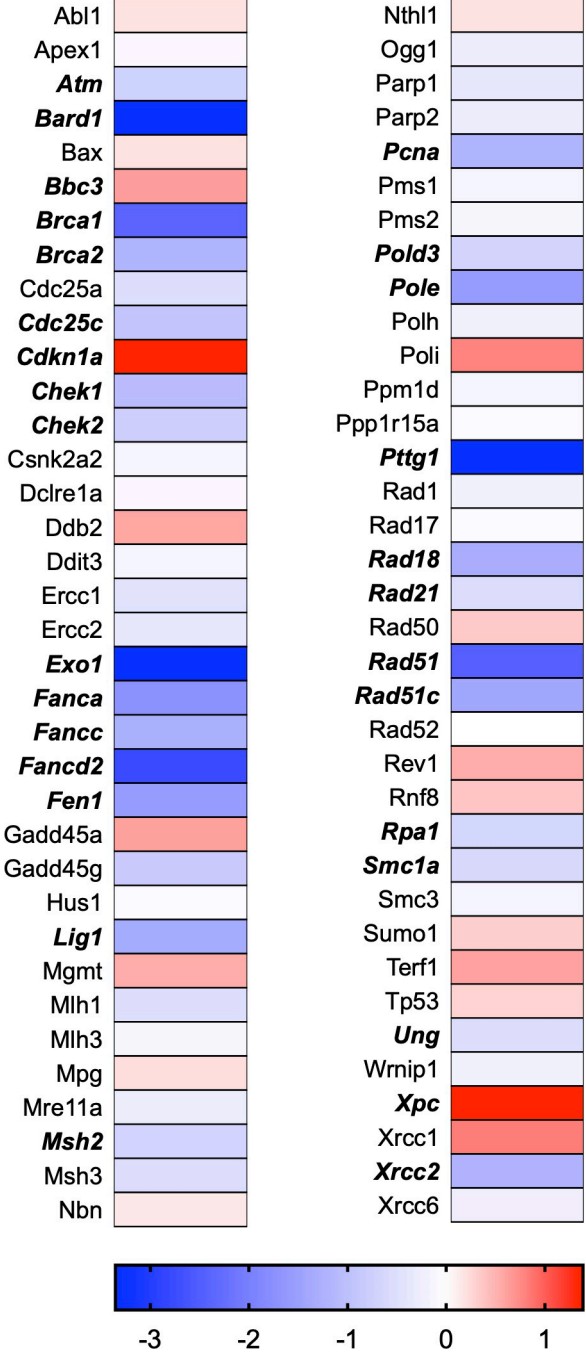

**Fig 4. DNA damage heat map.** Bold text indicates significantly altered gene targets.

descriptive approach to pathway analysis. We generated heat maps based on GeneGlobe RT$^2$ profiler arrays independent of whether those gene targets were significantly altered in our

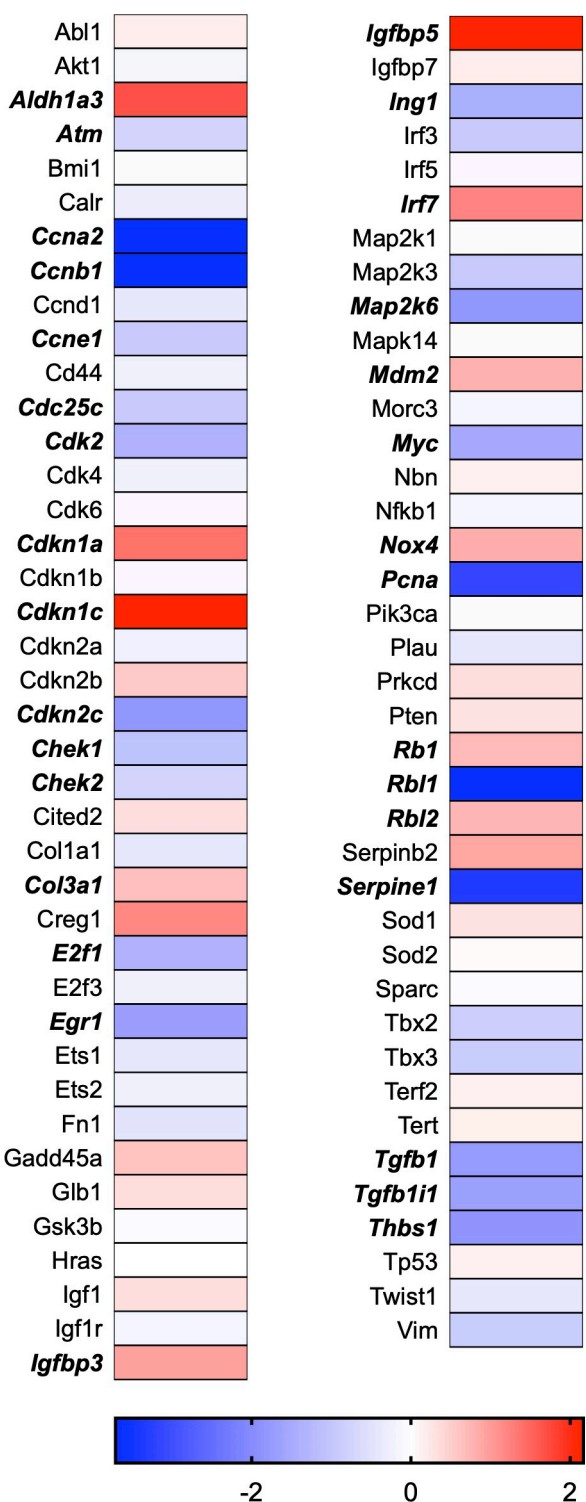

**Fig 5. Cellular senescence heat map.** Bold text indicates significantly altered gene targets.

dataset. Gene targets in the RT$^2$ profilers but not in our dataset were excluded from heat maps. The percentage of significantly altered genes, both increased and decreased, was calculated based on the number of total genes included in each pathway's respective heat map to rank the most implicated pathways. This systematic approach was employed to maintain an objective view of the global data.

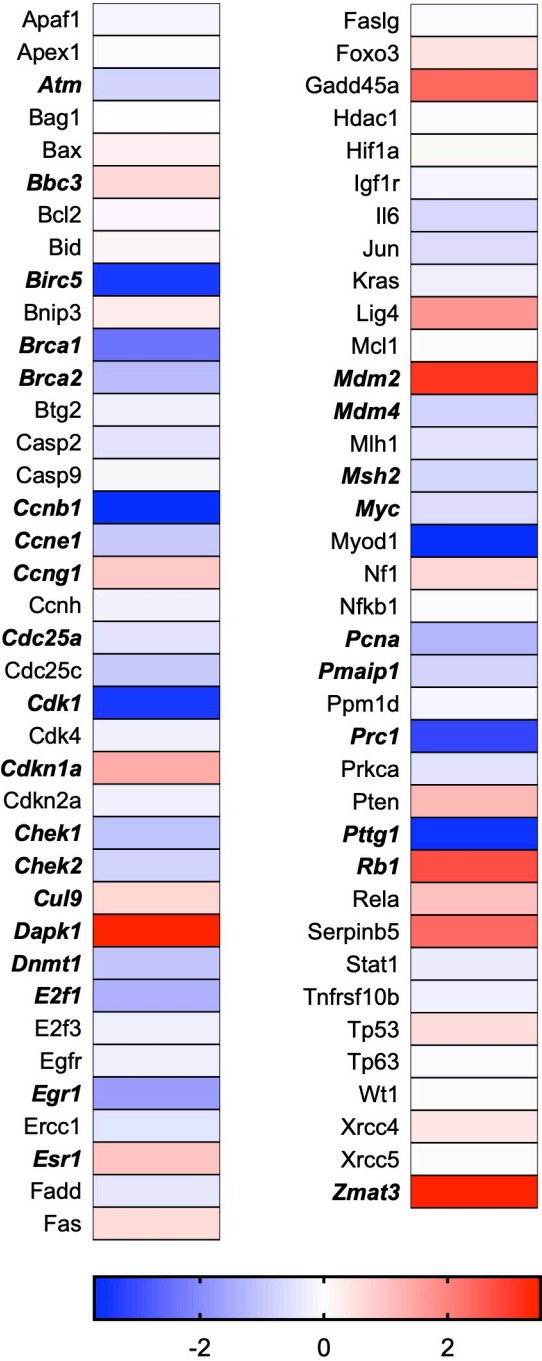

**Fig 6. p53 signaling heat map.** Bold text indicates significantly altered gene targets.

## Pathway analysis

RNAseq data were imported into Ingenuity Pathway Analysis (IPA, Qiagen) to determine select pathways and biological functions that were altered in response to AGE-BSA treatment.

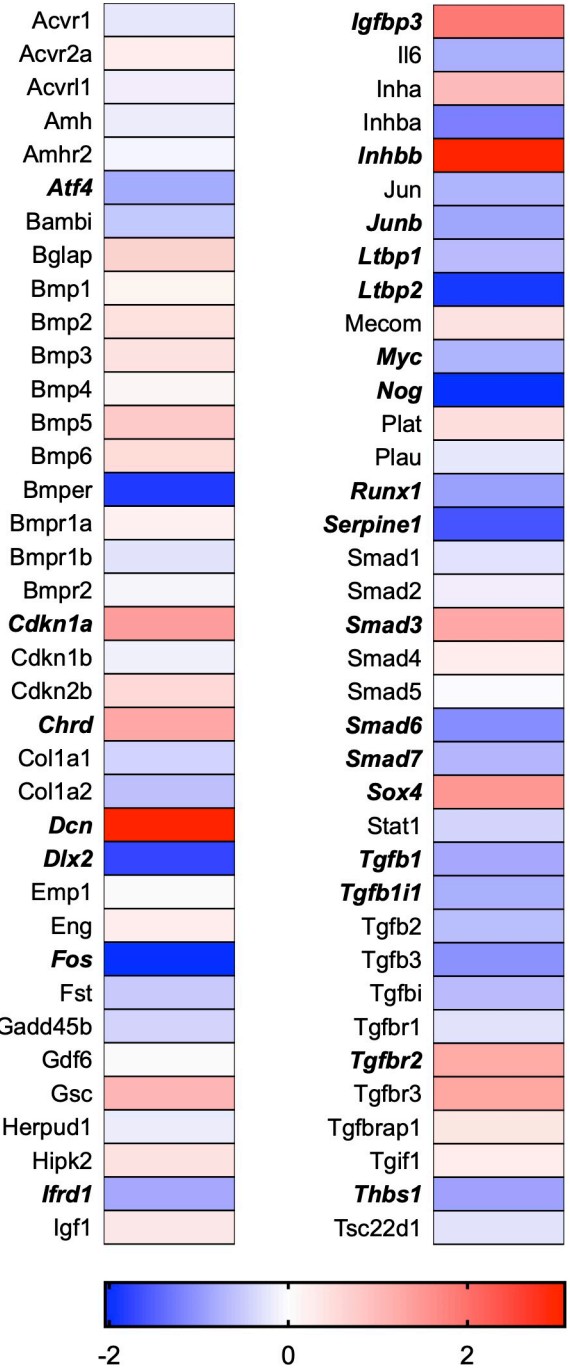

**Fig 7. TGF-β signaling heat map.** Bold text indicates significantly altered gene targets.

## Results and discussion

### Overview

A total of 2,159 genes within our data set met the criteria of q<0.05 and fold change of greater than or less than 1.5 ($\log_2$ fold change greater than or less than 0.584). One thousand forty-six genes were significantly increased, and 1,113 were significantly decreased (Fig 1).

### Most affected gene targets

The top ten increased, and the top ten decreased gene targets within our data set were identified based on our $\log_2$ fold change and adjusted p-value thresholds. The top ten increased gene

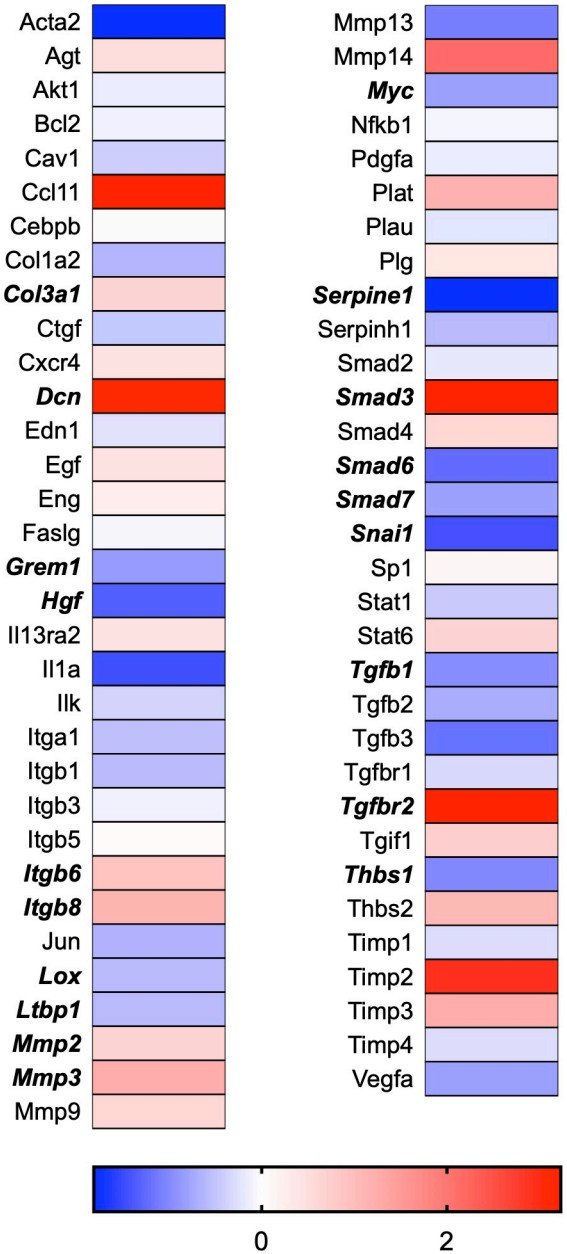

**Fig 8. Fibrosis heat map.** Bold text indicates significantly altered gene targets.

targets in order of highest to lowest positive $\log_2$ fold change were Cyp1a1, Pipox, Btc, Slc22a14, Tbxas1, Itgb2, Slc13a3, Cldn1, Ncf1, and Tnfrsf17 (Table 1). The top ten decreased gene targets in order of highest to lowest negative $\log_2$ fold change were Pimreg, Pmch, E2f7, Pbk, Parpbp, Ube2c, Troap, Cenpf, Cldn23, and Ccnb2 (Table 1).

## Descriptive pathway profiling

A total of 21 GeneGlobe (Qiagen) pathways were explored. Pathway selection was based on the literature, hypotheses that we have explored previously, and hypotheses we plan to explore in future studies. Select pathways strongly associated with AGE/RAGE biology were also included. Pathways were ranked strictly based on the percentage of significantly altered genes within that respective pathway. Pathways explored, in order from most to least

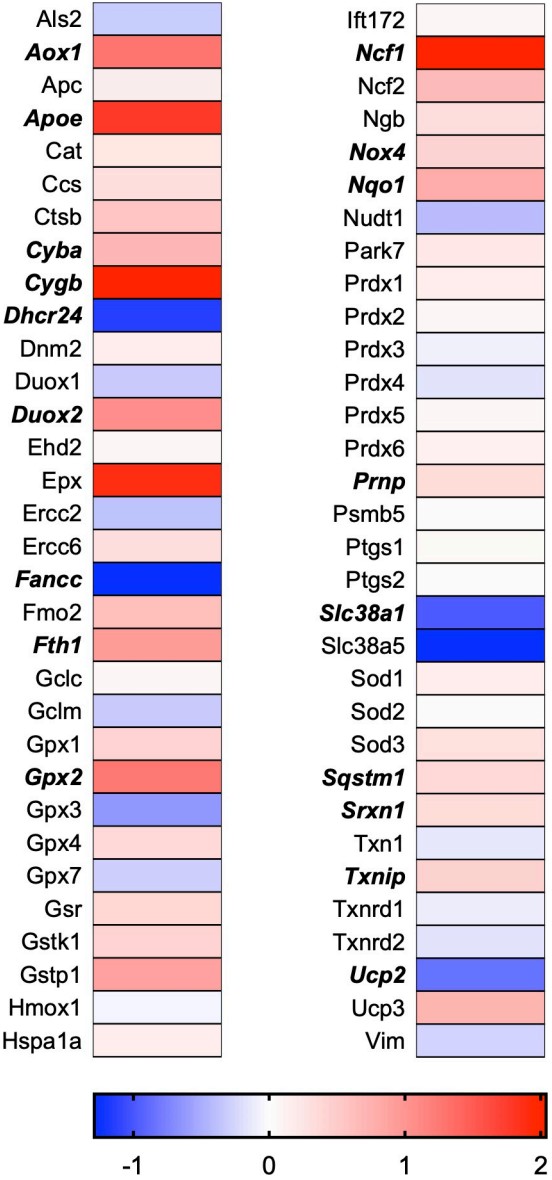

**Fig 9. Oxidative stress heat map.** Bold text indicates significantly altered gene targets.

implicated, were cell cycle (51.2%, Fig 2), extracellular matrix (ECM) and tenogenic markers (48.4%, Fig 3), DNA damage (40.3%, Fig 4), cellular senescence (39.2%, Fig 5), p53 signaling (38.7%, Fig 6), TGF-β signaling (32.4%, Fig 7), fibrosis (29.2%, Fig 8), oxidative stress (28.1%, Fig 9), wound healing (23.8%, Fig 10), growth factors (21.9%, Fig 11), transcription factors (20.6%, Fig 12), cytoskeleton (16%, Fig 13), cytokines (14.9%, Fig 14), innate and adaptive immunity (13.2%, Fig 15), NF-κB signaling (11.3%, Fig 16), cellular stress responses (10%, Fig 17), mitochondria (9.5%, Fig 18), apoptosis (8.5%, Fig 19), glycosylation (8.2%, Fig 20), inflammasomes (7.8%, Fig 21), and mitochondrial energy metabolism (2.6%, Fig 22). Pathways, listed in order of most implicated and respective figure numbers for heat maps, are summarized in Table 2.

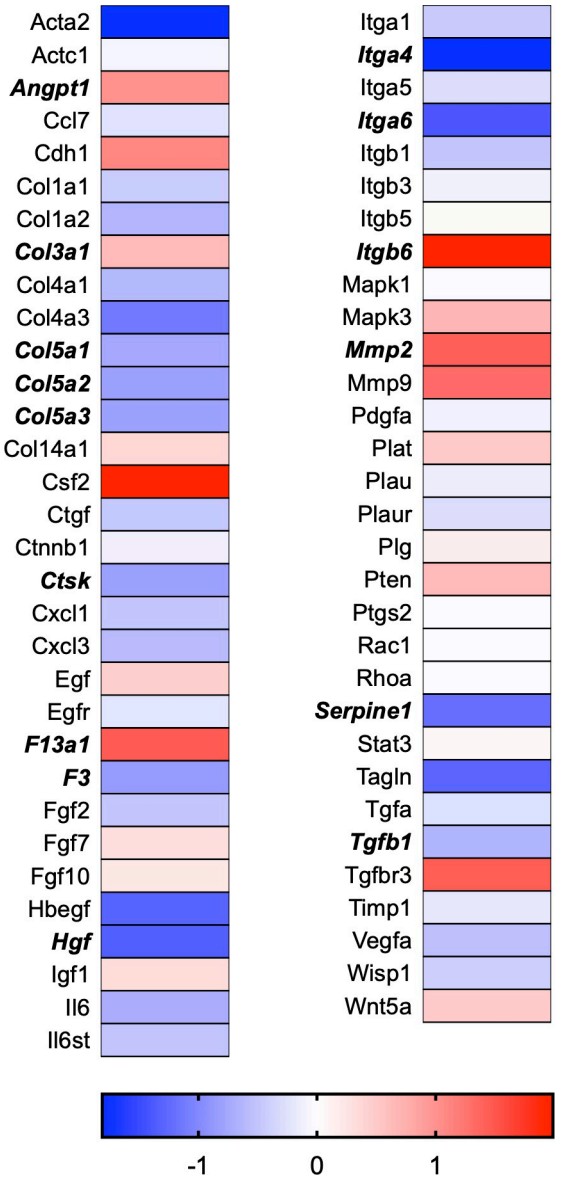

**Fig 10. Wound healing heat map.** Bold text indicates significantly altered gene targets.

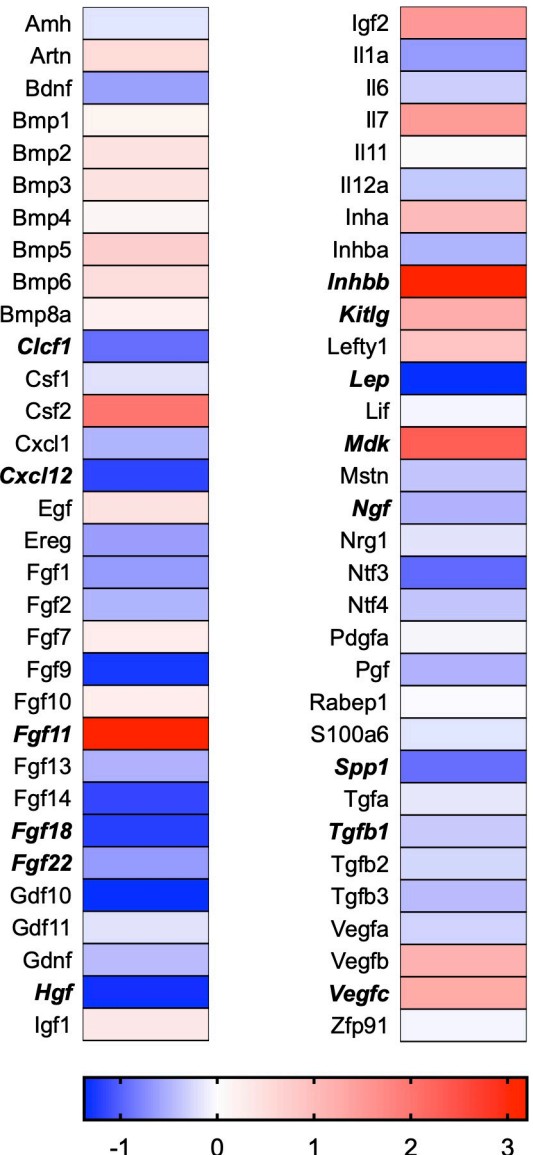

**Fig 11. Growth factors heat map.** Bold text indicates significantly altered gene targets.

## Pathway analysis

Ten pathways or biological functions were selected using the IPA disease and function tool. Apoptosis (Z Score: 4.70), morbidity and mortality (Z Score: 4.53), organismal death (Z Score: 4.47), DNA damage (Z Score: 3.36), and diabetes mellitus (Z Score: 2.24) were selected as activated pathways. Cell survival (Z Score: -4.91), cell viability (Z Score: -4.62), repair of DNA (Z Score: -3.85), cell proliferation (Z Score: -3.67), and growth of connective tissue (Z Score: -3.02) were selected as inhibited pathways. IPA pathways are summarized in Table 3 with respective p-values and activation Z-scores.

Diabetes-related complications, such as those implicating connective tissue, create a large healthcare burden and reduce quality of life. Our knowledge of diabetes-related tendon degeneration has primarily been limited to macroscopic and structural changes with minimal molecular insight exists. Previous work from our laboratory has demonstrated that AGEs induce

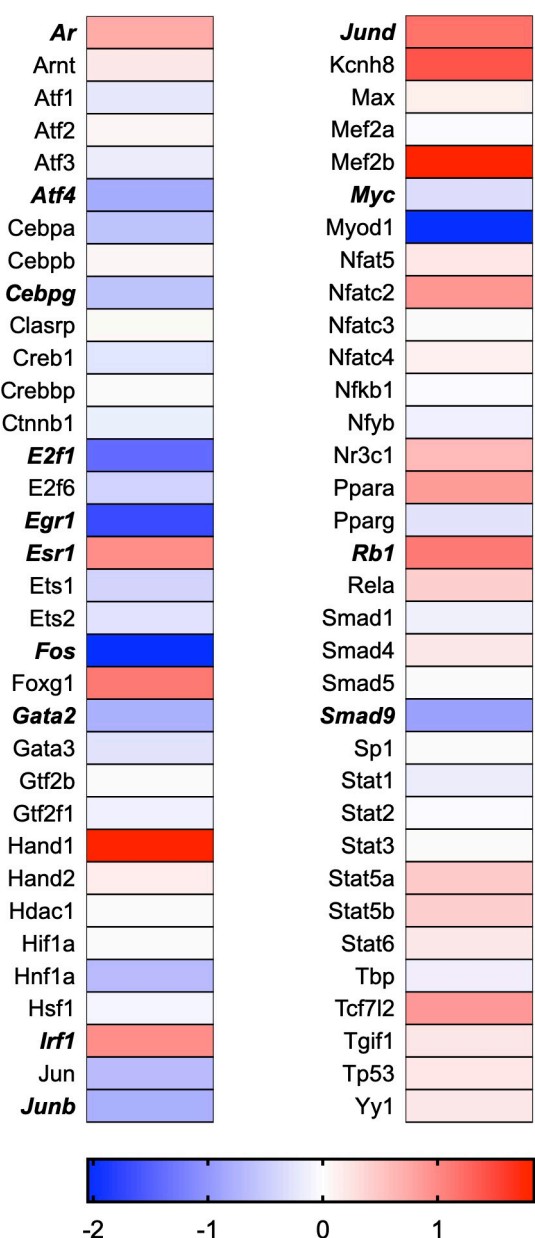

**Fig 12. Transcription factors heat map.** Bold text indicates significantly altered gene targets.

severe limitations to tendon fibroblast proliferative capacity and mitochondrial function while increasing mitochondrial DNA content [8]. We have followed up on these previous findings by completing a descriptive transcriptome profile of Achilles tendon-derived fibroblasts following AGE exposure. The goal of this study was to identify and rank pathways that were most implicated following AGE exposure, thus providing a more precise mechanistic exploration of AGE-mediated effects on tendon-derived cells.

Using a clinically-relevant concentration of AGEs [12, 22], we have previously demonstrated incorporation of synthetic nucleoside 5-ethynyl-2′-deoxyuridine (EdU) in tendon-derived fibroblasts to be ~3% following AGE-BSA (200μg/ml) exposure as compared to ~53% in the BSA-Control exposed group, which proliferate normally [8]. Further, we noted a

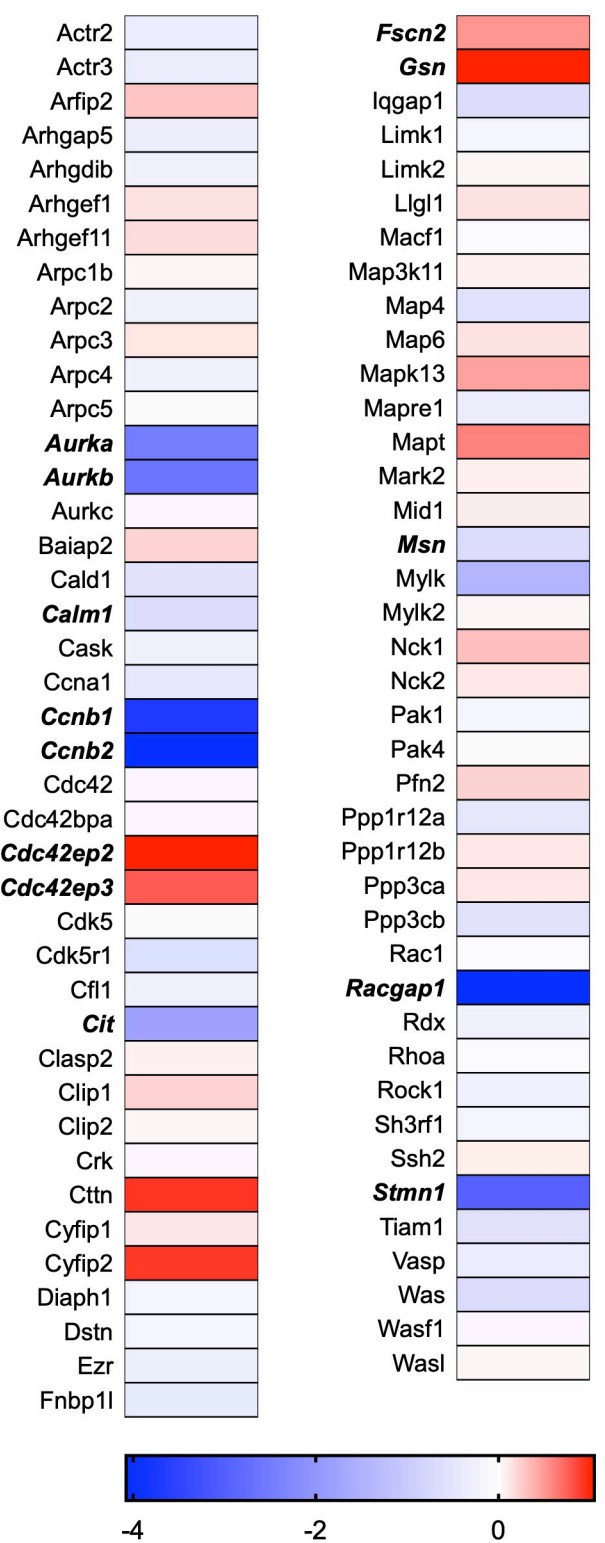

**Fig 13. Cytoskeleton heat map.** Bold text indicates significantly altered gene targets.

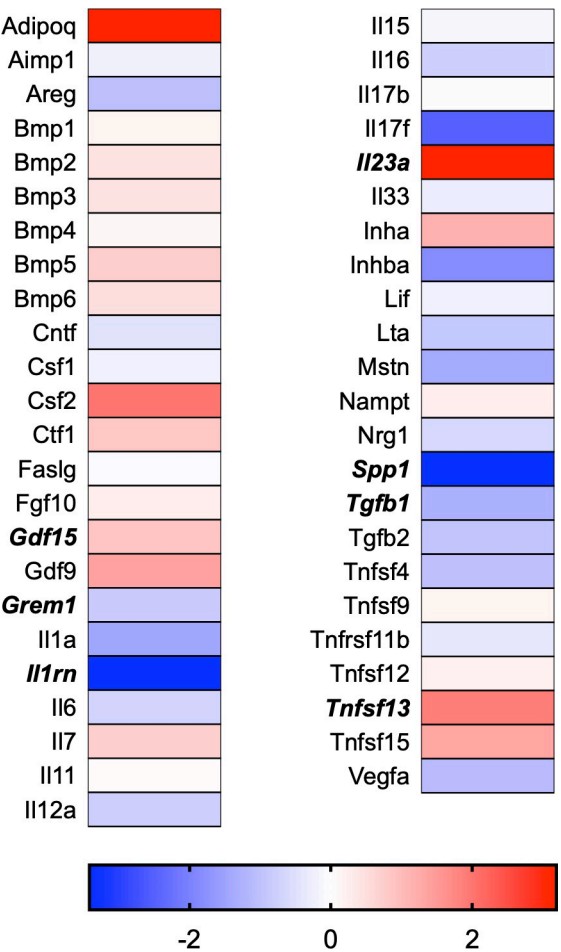

**Fig 14. Cytokines heat map.** Bold text indicates significantly altered gene targets.

reduction in proliferative gene markers, Mybl2 and Pcna, and reduced absorbance values of cytostatic MTT with AGE-BSA treatment in tendon fibroblasts. Our RNAseq data corroborated our previous findings of reduced Mybl2 and Pcna gene expression and revealed several additional genes responsible for cell cycle progression to be significantly impacted (Fig 2). In fact, our transcriptome analysis revealed that genes associated with the cell cycle are the most impacted by AGE treatment (Fig 2 and Table 2). Tendon fibroblast proliferation is vital for tendon development and adaptation [23, 24]. The inability of tenocytes to proliferate in the presence of AGEs could precipitate the development of tendon degeneration by limiting adaptations to loading [25]. Tendon healing requires a phase of increased cellular proliferation [23, 24, 26], thus AGE-induced limitations in cell proliferation could contribute to delayed in healing noted in those with diabetes [27–29]. In fact, would healing was identified as one of the top 10 GeneGlobe Pathways impacted by AGE treatment (Table 2 and Fig 10).

Gene targets associated with ECM maintenance and remodeling were also dramatically affected in our dataset (Fig 3). The ECM is vital to tendon tissue health and serves several vital functions, including cell adhesion, communication, and differentiation. Additionally, the ECM provides structural and biochemical support to the surrounding resident cell population. The tendon ECM consists primarily of type I and type III collagen fibers surrounded by proteoglycans that assist collagen fibrils' assembly and stability [30]. A precise and linear

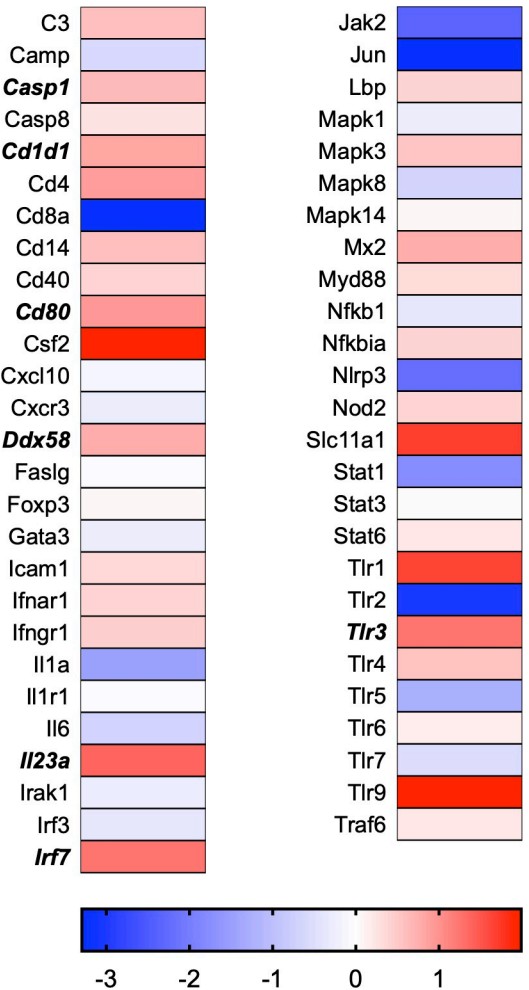

**Fig 15. Innate and adaptive immunity heat map.** Bold text indicates significantly altered gene targets.

arrangement of collagen fibrils is vital to tissue integrity and, therefore, mechanical function [31]. The inclusion of multiple collagen isoforms allows the ECM to specialize and adapt to specific mechanical loading and functional responses [32]. For instance, type I collagen (Col1a1) is a stronger collagen isoform. In contrast, type III collagen (Col3a1) is weaker and generally upregulated in the early stages of tissue remodeling following exercise or during the initial stages of healing [33, 34]. Col3a1 can provide temporary tensile strength to the tissue assembly until it is later replaced by stronger Col1a1 [35]. Although Col1a1 mRNA was unaffected in our RNAseq data set, Col3a1 mRNA expression was increased (Fig 3). Similarly, our previous report indicated Col3a1 mRNA expression increased with 50μg/ml and 100μg/ml AGE exposure compared to an equal dose of BSA-Control [8]. This increase in Col3a1 mRNA expression is likely in response to the AGE insult and an attempt to maintain the ECM environment.

Further, the most abundant tendon proteoglycan gene expression of decorin (Dcn) increased in our RNAseq data set (Fig 3). Dcn aids in the maintenance and regulation of collagen fibril structure and resident fibroblast proliferation [31]. As a critical regulator in matrix assembly, loss of Dcn would likely prove to be unfavorable to the strength of the tendon assembly, which would decrease the tissue's ability to withstand sudden strain [31]. Our observed

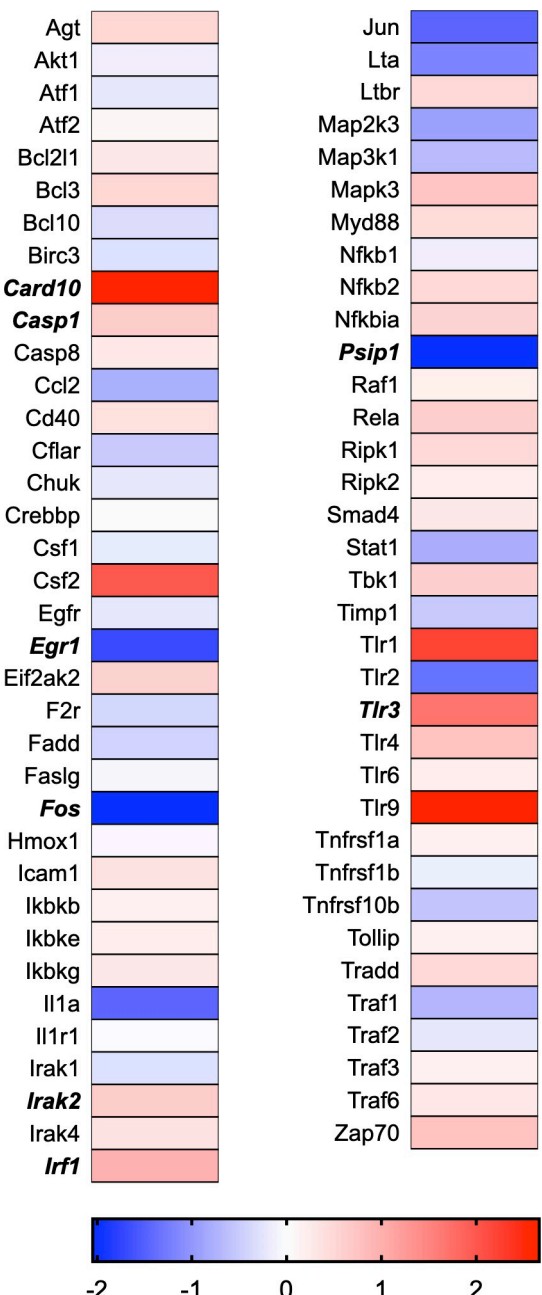

**Fig 16. NF-κB signaling heat map.** Bold text indicates significantly altered gene targets.

increase in Dcn gene expression may be a compensatory response that results in response to the AGE insult. However, impacts to Dcn content and gene expression would need to be externally validated in a whole diabetic tendon.

Lysine and hydroxylysine are found within the collagen amino acid sequence and play an essential role in cross-link formation. Oxidation of lysine and hydroxylysine by lysyl oxidase (Lox) forms cross-links within collagen fibrils, contributing to tissue integrity by increasing tensile strength and stabilizing the collagen fibril assembly. Strength and stability of the tissue assembly are essential, especially given the high contractile forces tendons

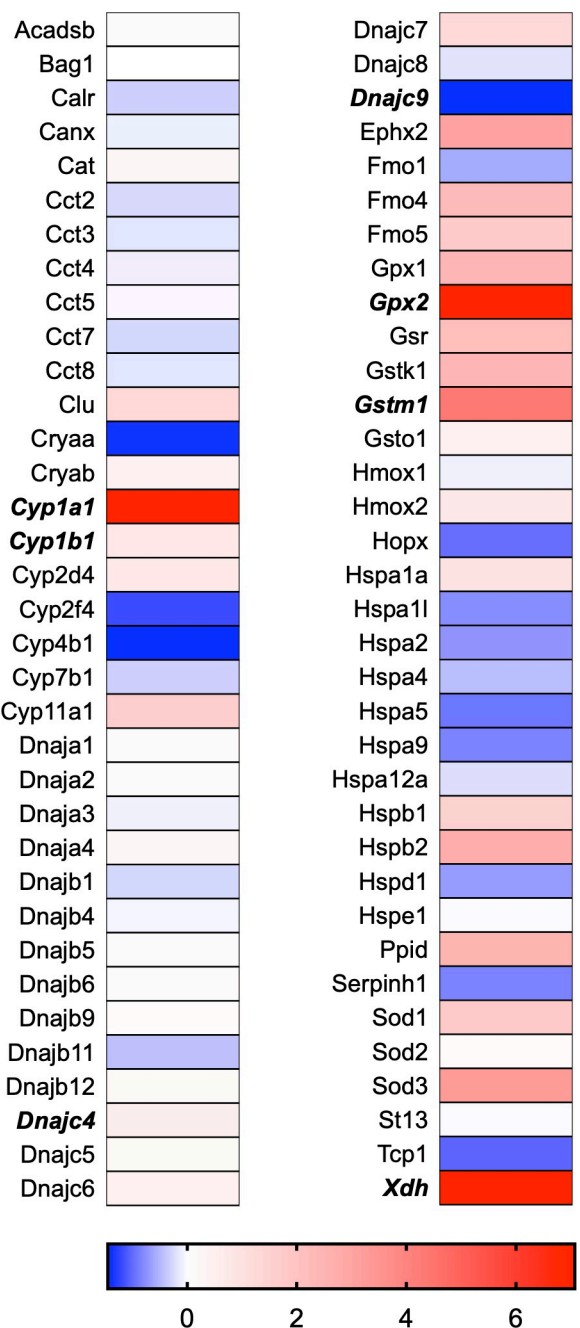

**Fig 17. Cellular stress responses heat map.** Bold text indicates significantly altered gene targets.

are responsible for transmitting from muscle to bone. Our dataset revealed Lox gene expression to be significantly reduced following AGE exposure (Fig 3). If reduced mRNA expression of Lox coincides with reduced enzymatic cross-link formation, AGEs may contribute to a weakened tendon assembly due to loss of enzymatic cross-links between adjacent collagen fibrils. Tendons of diabetic animals generally have a reduced load to failure capacity, which may be a result of greater tissue degeneration at the macroscopic level [4, 11, 28, 36]. More work is needed to determine the impact of AGEs on the whole tendon fibril assembly.

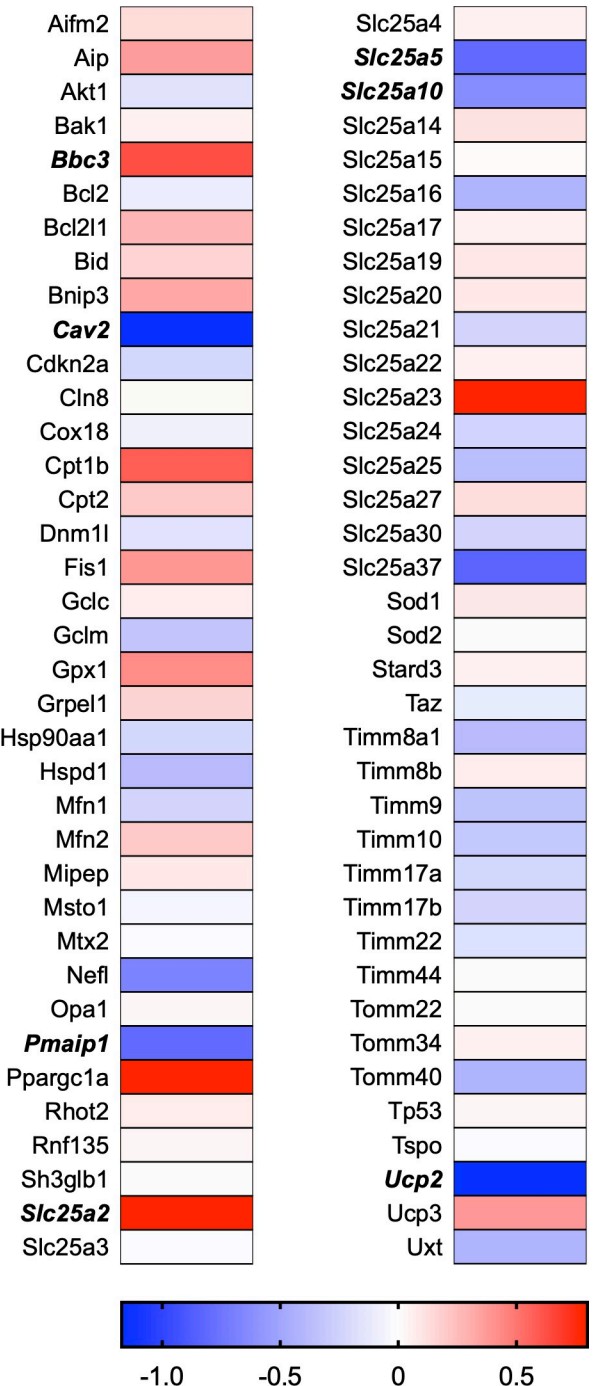

**Fig 18. Mitochondria heat map.** Bold text indicates significantly altered gene targets.

Remodeling of the ECM is primarily regulated by enzymes known as matrix metallopro-teinases (MMPs), which are responsible for the degradation portion of ECM remodeling. Collagenases such as MMP-1 and MMP-13 cleave type I collagen molecules in the ECM. Similarly, gelatinases, such as MMP-2 and MMP-9, degrade collagen isoforms in the ECM. MMPs are transcribed and translated as proenzymes and then secreted into the ECM,

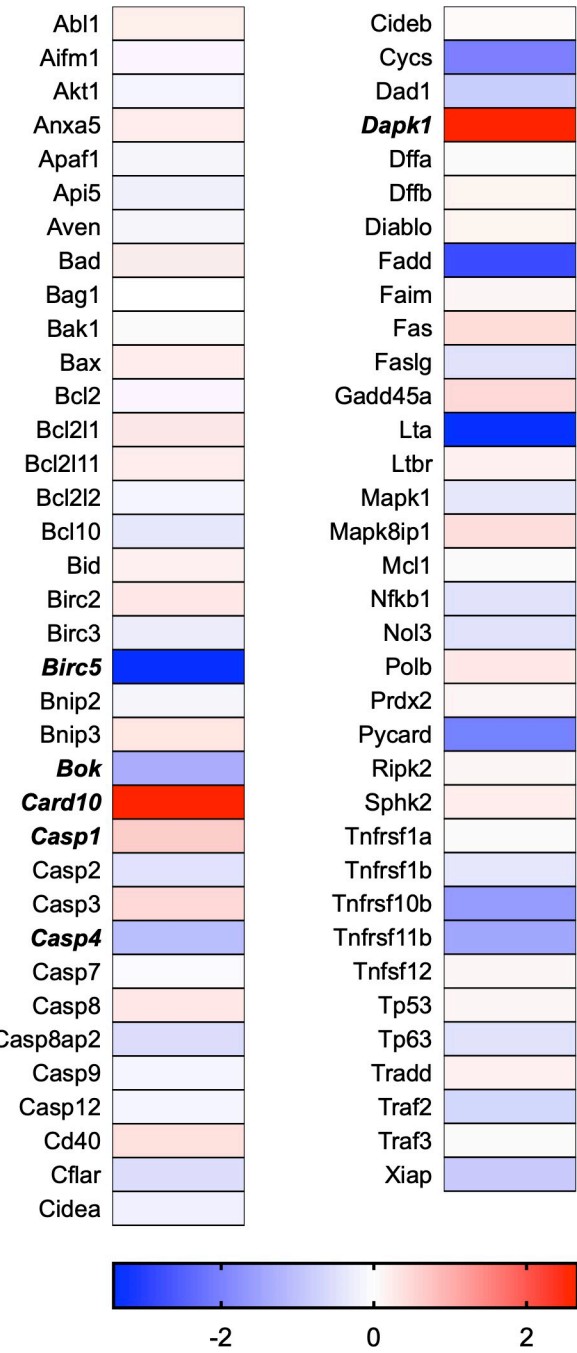

**Fig 19. Apoptosis heat map.** Bold text indicates significantly altered gene targets.

where they are activated through proteolytic cleavage of the N-terminal. Although MMP activity is degenerative, it facilitates ECM remodeling and tendon tissue adaptation. In turn, MMP activity can be reversibly inhibited by a group of enzymes known as tissue inhibitors of metalloproteinases (TIMPs). TIMPs play an essential role in ECM remodeling by limiting MMP activity and preventing excessive degradation. Counter-regulation via TIMP activity tightly regulates the breakdown and synthesis of collagen in response to external stresses,

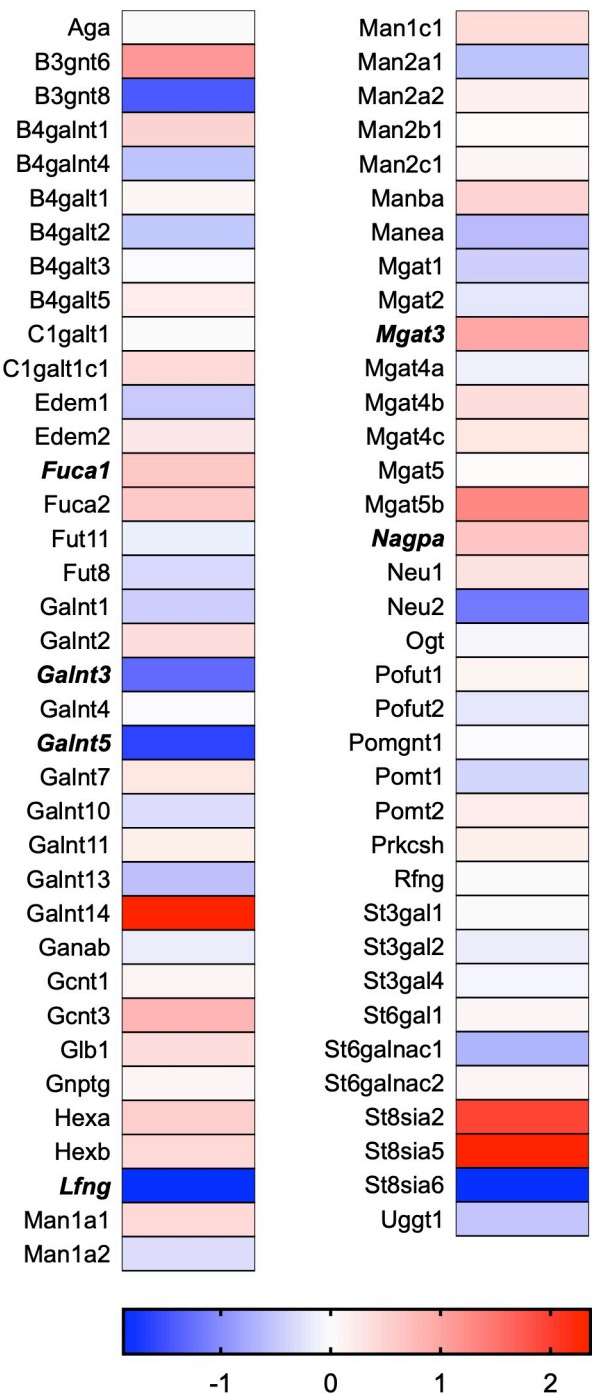

**Fig 20. Glycosylation heat map.** Bold text indicates significantly altered gene targets.

such as mechanical loading. Loss of ECM regulation, such as favoring degradation over synthesis, could alter the ECM responses to damage the tissue assembly. It is no surprise that the dysregulation of degenerative enzymes, such as MMPs, has been thought to play an essential role in developing tendon pathology in diabetes as overexpression of MMPs may favor ECM degradation [37]. Similarly, if inhibitory TIMPs are less expressed, the

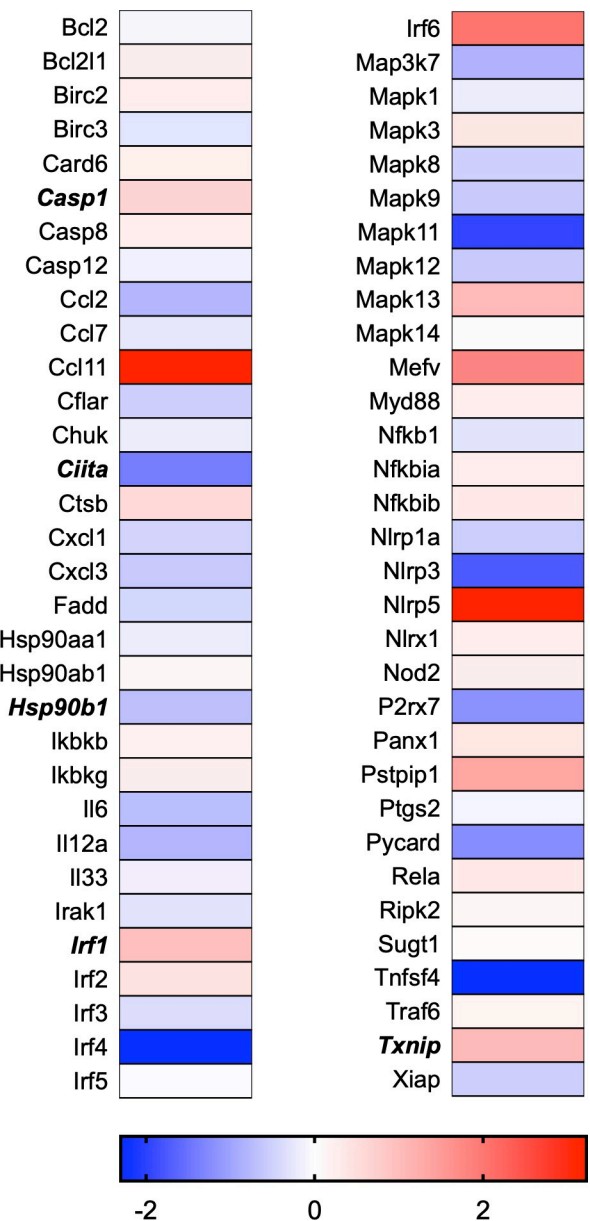

**Fig 21. Inflammasomes heat map.** Bold text indicates significantly altered gene targets.

environment may also favor degradation by allowing MMPs to act on the ECM for a more extended period. Previous reports have indicated that AGEs increase MMP -2, -3, -9, and -13 secretion and expression in chondrocytes with 100μg/ml of AGEs [38, 39]. Further, mRNA expression of MMP -1, -3, and -13 in porcine chondrocytes was increased with 100μg/ml of AGE exposure [40]. Our previous work in Achilles tendon-derived fibroblasts demonstrated an increase in MMP -2 and -3 but no significant changes to MMP-9 and -13 [8]. Our RNAseq analysis confirmed MMP -2 and -3 to be elevated, along with MMP -15 and -17. However, we did not observe any changes to TIMP -1, -2, -3, or -4 in our RNAseq dataset, suggesting that MMPs may be exerting their function in an unorganized fashion that would favor a degenerative ECM environment. MMP gene expression data is limited in

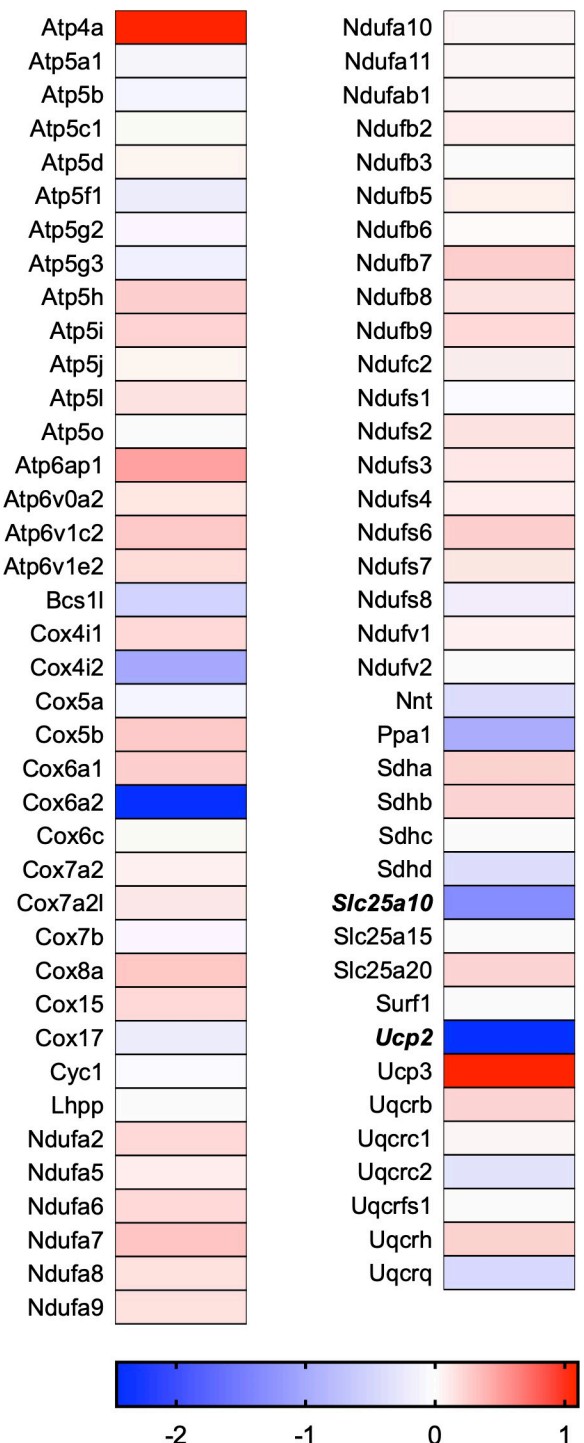

**Fig 22. Mitochondrial energy metabolism heat map.** Bold text indicates significantly altered gene targets.

scope as it does not account for ECM secretion and N-terminal cleavage. However, the large impact that AGE exposure has on the dysregulation of ECM-related gene expression is further evidence that elevated serum AGEs may be contributing to the development of connective tissue pathology in diabetic populations (Fig 3).

**Table 2. Descriptive pathway profiling.**

| Figure | GeneGlobe Pathway | Altered Genes in Pathway | Total Genes in Pathway | Percent of Affected Genes |
|---|---|---|---|---|
| 2 | Cell Cycle | 42 | 82 | 51.2 |
| 3 | ECM and Tenogenic Markers | 31 | 64 | 48.4 |
| 4 | DNA Damage | 29 | 72 | 40.3 |
| 5 | Cellular Senescence | 31 | 79 | 39.2 |
| 6 | p53 Signaling | 29 | 75 | 38.7 |
| 7 | TGF-β Signaling | 24 | 74 | 32.4 |
| 8 | Fibrosis | 19 | 65 | 29.2 |
| 9 | Oxidative Stress | 18 | 64 | 28.1 |
| 10 | Wound Healing | 15 | 63 | 23.8 |
| 11 | Growth Factors | 14 | 64 | 21.9 |
| 12 | Transcription Factors | 14 | 68 | 20.6 |
| 13 | Cytoskeleton | 13 | 81 | 16 |
| 14 | Cytokines | 7 | 47 | 14.9 |
| 15 | Innate and Adaptive Immunity | 7 | 53 | 13.2 |
| 16 | NF-κB Signaling | 8 | 71 | 11.3 |
| 17 | Cellular Stress Responses | 7 | 70 | 10 |
| 18 | Mitochondria | 7 | 74 | 9.5 |
| 19 | Apoptosis | 6 | 71 | 8.5 |
| 20 | Glycosylation | 6 | 73 | 8.2 |
| 21 | Inflammasomes | 5 | 64 | 7.8 |
| 22 | Mitochondrial Energy Metabolism | 2 | 77 | 2.6 |

Delayed and abnormal healing is a common complication of types I and II diabetes [27, 41]. Not only does it appear that diabetic patients are at risk of developing tendon tears, but healing post-repair is also impaired [42–44]. Interestingly, transforming growth factor (TGF) β1 expression was significantly reduced in our RNAseq data (Fig 7). In addition to TGFβ1 being one of the affected genes in the wound-healing pathway (Fig 10), the GeneGlobe TGFβ signaling pathway was also strongly influenced by AGE treatment (Table 2 and Fig 7). TGFβ is a critical factor in fibrosis and modulation of ECM homeostasis [45]. It has previously been demonstrated that TGFβ levels are significantly reduced in diseased human rotator cuff tendon samples [45]. In addition, TGFβ is known to modulate inflammatory responses by influencing fibroblast recruitment and stimulating collagen production [46, 47].

**Table 3. Select IPA pathway analysis.**

| Pathway | p Value | Activation Z Score |
|---|---|---|
| Apoptosis | 1.45E-33 | 4.70 |
| Morbidity or Mortality | 4.62E-34 | 4.53 |
| Organismal Death | 2.06E-33 | 4.47 |
| DNA Damage | 7.32E-09 | 3.36 |
| Diabetes Mellitus | 1.27E-13 | 2.24 |
| Cell Survival | 4.74E-25 | -4.91 |
| Cell Viability | 5.59E-23 | -4.62 |
| Repair of DNA | 7.15E-15 | -3.85 |
| Cell Proliferation (Fibroblast) | 7.59E-12 | -3.67 |
| Growth of Connective Tissue | 5.10E-23 | -3.02 |

Inconsistent with known effect of TGFβ on collagen production [46, 47], Col1a1 was unchanged in our RNAseq dataset, and Col3a1 was increased (Fig 3). However, mRNA expression of Col5a1, Col5a2, and Col5a3 expression was significantly reduced in our RNAseq dataset. Type V is a fibrillar collagen isoform found less abundantly in a tendon but exists to provide support to tissues that do contain high levels of type V collagen isoforms [48]. While the wound healing GeneGlobe pathway was not as affected as other pathways, it is still likely that these gene targets contribute in some manner to the delayed healing response that is commonly observed following tendon injury in diabetic patients.

## Conclusions

Several studies have shown that the risk of developing tendinopathy is greater in those with diabetes mellitus [42–44, 49]. Our new data highlights cell-signaling pathways that may assist with expanding our understanding of diabetic tendon pathology and failed healing responses. While our discussion is limited in scope, and we provide only transcriptome data, the purpose of this study was to complete a descriptive profile of the AGE insult to tendon fibroblasts. This work is the first data set to utilize RNAseq methodology to study the tendon fibroblast transcriptome following AGE exposure. These data will be helpful for further elucidation of the diabetic tendon disease process.

## Author Contributions

**Conceptualization:** Shivam H. Patel, Chad C. Carroll.

**Data curation:** Shivam H. Patel, Chad C. Carroll.

**Formal analysis:** Shivam H. Patel, Christopher L. Mendias, Chad C. Carroll.

**Funding acquisition:** Shivam H. Patel, Chad C. Carroll.

**Investigation:** Shivam H. Patel, Chad C. Carroll.

**Methodology:** Shivam H. Patel, Christopher L. Mendias, Chad C. Carroll.

**Project administration:** Shivam H. Patel, Chad C. Carroll.

**Resources:** Chad C. Carroll.

**Software:** Christopher L. Mendias.

**Supervision:** Christopher L. Mendias, Chad C. Carroll.

**Validation:** Shivam H. Patel, Chad C. Carroll.

**Visualization:** Shivam H. Patel, Christopher L. Mendias.

**Writing – original draft:** Shivam H. Patel, Chad C. Carroll.

**Writing – review & editing:** Shivam H. Patel, Chad C. Carroll.

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
