## [Decision Letter · Decision Letter 0]

26 Apr 2022

PONE-D-22-07074Descriptive Transcriptome Analysis of Tendon Derived Fibroblasts Following In-Vitro Exposure to Advanced Glycation End ProductsPLOS ONE

Dear Dr. Carroll,

Thank you for submitting your manuscript to PLOS ONE. After careful consideration, we feel that it has merit but does not fully meet PLOS ONE’s publication criteria as it currently stands. Therefore, we invite you to submit a revised version of the manuscript that addresses the points raised during the review process.

We look forward to receiving your revised manuscript.

Kind regards,

Ming-Chang Chiang

Academic Editor

PLOS ONE

Journal Requirements:

"This work was supported by NIH pre-doctoral fellowship F31-AR073647 (S.H.P), Purdue University Research Initiative Funds (C.C.C), and Ralph W. and Grace M. Showalter Research Trust Award (C.C.C.)."

"This work was supported by NIH pre-doctoral fellowship F31-AR073647 (S.H.P), Purdue University Research Initiative Funds (C.C.C), and Ralph W. and Grace M. Showalter Research Trust Award (C.C.C.). The sponsors did not have a role in the study design, data collection and analysis, decision to publish, or preparation of the manuscript"

Reviewers' comments:

Reviewer's Responses to Questions

**Comments to the Author**

1. Is the manuscript technically sound, and do the data support the conclusions?

Reviewer #1: Yes

Reviewer #2: Yes

Reviewer #3: Yes

2. Has the statistical analysis been performed appropriately and rigorously? 

Reviewer #1: Yes

Reviewer #2: Yes

Reviewer #3: I Don't Know

3. Have the authors made all data underlying the findings in their manuscript fully available?

Reviewer #1: Yes

Reviewer #2: No

Reviewer #3: Yes

4. Is the manuscript presented in an intelligible fashion and written in standard English?

Reviewer #1: Yes

Reviewer #2: Yes

Reviewer #3: Yes

5. Review Comments to the Author

Reviewer #1: I congratulate authors to extend their previous study with a comprehensive gene profiling in rat tendon derived fibroblasts. AGEs exposure mainly affects pathways involved cell survival and proliferation as well as ECM remodeling. The data help to understand diabetic tendinopathy mechanisms to ask whether AGEs are the causal for the complication.

I do have a critique that authors used only one concentration of AGEs. I understand authors picked up the concentration from previous study. Cell death and damage pathways were strongly shown up in the analysis. I am worried that too severe damages can be a confounding factor to evaluate transcriptome changes. Meaning, mechanical stress inducing cell death may cause all the transcriptome changes. Are authors sure that the AGE concentration used here is clinically relevant concentration compared to diabetes patient serum? These discussions should be included in the manuscript.

Another minor points are:

• Please add GEO accession number if it is available in cover pages.

• Please spell out AGE-BSA vendor and catalog number in method section. AGEs consist of heterogenic products. Authors want to make the study reproducible for others.

• Ingenuity pathway analysis enables to determine upstream pathways. Authors may mitigate the question above too much AGEs mediated cell death.

Reviewer #2: In this manuscript, Patel et al. analyzed the transcriptome change of rat tendon-derived fibroblasts treated with AGEs. The top 21 implicated cell-signaling pathways are described and show that AGEs disrupt the transcriptome of tendon fibroblasts on a large scale, which may have some implications for future research in diabetic tendinopathy. However, due to the lack of further experimental verification and the small sample size of RNAseq, this study is of little significance. I think the authors should have further studies in this manuscript based on the RNAseq results rather than just analyzing the RNAseq results of a few samples.

In addition, I also have the following comments/suggestions for the authors’ consideration.

1. The sentence in the results section: “A total of 2,159 genes within our data set met the criteria of q<0.05 and fold change of greater than or less than 1.5 (log2 fold change greater than or less than 0.584)” needs clarification. The “q” is p-value, right? This problem also arises in Figure 1. What does “fold change of greater than or less than 1.5” mean?

2. The gene Cyp1a1 (log2 Fold Change is 7.07) does not show in Figure 1. What gene does the point on about log2 Fold Change 5 of Figure 1 represent? The gene names in Table1 should be shown in Figure 1.

3. The data in the table1 should be discussed in the discussion section.

4. The expression level of some genes, which were significant differences in RNAseq results and mentioned in the discussion section, should be re-validated by another experimental technique, such as QPCR.

5. In implicated cell-signaling pathways, network analysis of dysregulated genes needs to be supplemented to clarify the interaction of the genes.

6. The first paragraph of the discussion section repeats a lot of information already presented in the introduction section. These should be shortened.

7. The first ECM should be given its full name.

8. Some studies have shown that the expression level of Tnmd is decreased, but the expression level of this gene is increased in Figure3, why? I think this data needs to be verified again.

9. Some unimportant Figures and Table should be attached in Supplementary Information. The original sequencing results should also be uploaded.

Reviewer #3: This a beneficial study that investigate transcriptome-wide changes that occur to

tendon-derived fibroblasts following treatment with AGEs. There were some doubts about the number of rats in treated (exposed) and control and need clarification. Several varieties were evaluated that showed in 22 figures. Except numerous number of figures, I didn’t find any fault in the manuscript.

6. PLOS authors have the option to publish the peer review history of their article (what does this mean?). If published, this will include your full peer review and any attached files.

Reviewer #1: **Yes: **Yuichiro Adachi

Reviewer #2: No

Reviewer #3: No

---

## [Author Response · Author response to Decision Letter 0]

28 Jun 2022

Reviewer 1

I congratulate the authors on extending their previous study with a comprehensive gene profiling in rat tendon-derived fibroblasts. AGEs exposure mainly affects pathways involved in cell survival and proliferation, and ECM remodeling. The data help understand diabetic tendinopathy mechanisms to ask whether AGEs are the causal for the complication. 

I do have a critique that the authors used only one concentration of AGEs. I understand the authors picked up the concentration from the previous study. Cell death and damage pathways were strongly shown up in the analysis. I am worried that too severe damages can be a confounding factor to evaluate transcriptome changes. Meaning, mechanical stress inducing cell death may cause all the transcriptome changes. Are authors sure that the AGE concentration used here is clinically relevant concentration compared to diabetes patient serum? These discussions should be included in the manuscript. 

Our initial report established a clear dose-response relationship between AGE concentration and effects on tendon cells. Our dose-response data from 0-200 �g/ml) suggest no inflection point. All doses produce a similar direction in gene changes. We believe that the strong effects of AGEs on cells only strengthen the need further to determine the long-term impact of AGEs on tendon properties and explore the mechanisms of these effects. Including multiple doses would confound the RNA sequencing analysis and would like not to reveal any additional information, i.e., we would see similar trends in pathway activation/suppression just to a lower magnitude. Cells were not mechanically stressed, and the concentration used fits within the range of AGEs concentrations seen in an individual with diabetes (1, 2).

Another minor points are:

• Please add GEO accession number if it is available in cover pages. • Please spell out AGE-BSA vendor and catalog number in method section. AGEs consist of heterogenic products. Authors want to make the study reproducible for others.

• Ingenuity pathway analysis enables to determine upstream pathways. Authors may mitigate the question above too much AGEs mediated cell death. 

1. We have uploaded our data to GEO (GSE204714).

2. The AGE-BSA is made “in-house,” as we have reported previously. (3) We have included additional details in the Methods and materials. Thank you.

3. In our previous data, 50 ug/ml was sufficient to induce limitations to cell proliferation and mitochondrial function. However, to maintain external validity, we choose a dose clinically relevant in the context of type 2 diabetes.

 

Reviewer 2

1. In this manuscript, Patel et al. analyzed the transcriptome change of rat tendon-derived fibroblasts treated with AGEs. The top 21 implicated cell-signaling pathways are described and show that AGEs disrupt the transcriptome of tendon fibroblasts on a large scale, which may have some implications for future research in diabetic tendinopathy. However, due to the lack of further experimental verification and the small sample size of RNAseq, this study is of little significance. I think the authors should have further studies in this manuscript based on the RNAseq results rather than just analyzing the RNAseq results of a few samples.

We thank the reviewer for their comments but respectfully disagree on the issue with sample size. It's challenging to determine a required sample size a priori for in vitro RNAseq studies. However, a general approach is to take a minimum sample size of N=4 and evaluate the coefficient of variation of key genes thought to be central to this process. Based on previous RNAseq studies using cultured tendon fibroblasts (4, 5), a sample size of N=4 provides low coefficients of variation and is sufficient to detect differences in key tenogenic, cell proliferation, and apoptotic genes, changes that match observations in the whole tendon. In this study, we used N=5 for each group and therefore feel we have an adequate sample size to test the central hypotheses of this manuscript. Further, studies have shown that 15 million reads (we used 40 million in our study) are adequate for power (6) and a sample size of five will yield sufficient statistical power (7).

2. The sentence in the results section: “A total of 2,159 genes within our data set met the criteria of q<0.05 and fold change of greater than or less than 1.5 (log2 fold change greater than or less than 0.584)” needs clarification. The “q” is p-value, right? This problem also arises in Figure 1. What does “fold change of greater than or less than 1.5” mean?

We used a very standard approach for RNAseq experimentation and data analysis based on the guidance of the bioinformatics experts in our genomics core. The q value is the false discovery rate (FDR) adjusted p-value. An FDR correction is applied to adjust for the multiple observations made. Setting a requirement for a gene to have a q-value < 0.05 and a 1.5-fold change difference before it is considered significantly different is standard. Because the volcano plot graphs are displayed on a log scale, we log transform the 1.5-fold difference (log2 1.5 = 0.584).

3. The gene Cyp1a1 (log2 Fold Change is 7.07) does not show in Figure 1. What gene does the point on about log2 Fold Change 5 of Figure 1 represent? The gene names in Table1 should be shown in Figure 1.

Thank you for the comments. Any x-y data point outside the boundary of the volcano plot is represented directly on the y axis. We included Table 1 to highlight the top gene changes. The volcano plot functions as a method to visualize the overall spread of the RNA sequencing data, and the tables highlight individual changes in gene expression. 

4. The data in the table1 should be discussed in the discussion section.

To preserve unbiased gene target selection and maintain a hypothesis-driven pathway selection, GeneGlobe (Qiagen, Hilden, Germany) pathway database was utilized to complete a descriptive approach to pathway analysis. Our study aimed to complete a pathway analysis of genes that were relevant based on our previous data and knowledge of diabetic tendon pathology. The top 10 altered genes are provided to establish the overall impact of AGEs on fibroblast for reference for future studies. 

5. The expression level of some genes, which were significant differences in RNAseq results and mentioned in the discussion section, should be re-validated by another experimental technique, such as QPCR.

We don't feel that qPCR is a necessary validation step for RNAseq. We used standard Illumina reagents and supplies that have been used in hundreds of thousands of previous studies. We have previously performed RNAseq in several tendon studies, and qPCR validation for these studies provided no different results than what was observed in RNAseq. Further, genes that we have assessed previously after AGE treatment in our cell culture model (3) followed similar patterns in our RNA sequencing studies. qPCR could be useful for looking at spliced isoform versions of specific genes where specific exon boundaries may not be picked up in all sequencing runs. But in the absence of this, we do not feel there is any additional value in performing qPCR experiments. 

6. In implicated cell-signaling pathways, network analysis of dysregulated genes needs to be supplemented to clarify the interaction of the genes.

We used the IPA disease and function tool for bioinformatics analysis instead of direct gene network analysis. We felt this approach would help us extrapolate the in vitro data to provide better translational insight into the in vivo physiology of diabetic tendon disease. 

7. The first paragraph of the discussion section repeats a lot of information already presented in the introduction section. These should be shortened.

Thank for the feedback. We have shortened this section of the Discussion.

8. The first ECM should be given its full name.

Thank you. We have made this correction.

9. Some studies have shown that the expression level of Tnmd is decreased, but the expression level of this gene is increased in Figure3, why? I think this data needs to be verified again.

The difference between Tnmd in our paper and others may be due to differences in experimental conditions. In previous papers, we have frequently measured Tnmd in qPCR and RNAseq, as discussed above, and have found the qPCR data to match the RNAseq data. Therefore, we are confident that the presented data adequately measures Tnmd expression. We are also unaware of any previous reports in tendon tissue or tendon cells assessing Tnmd expression after AGE exposure.

10. Some unimportant Figures and Table should be attached in Supplementary Information. The original sequencing results should also be uploaded.

We have moved several figures to Supplementary information. We have also uploaded our data to GEO.

Reviewer 3

This a beneficial study that investigated transcriptome-wide changes that occur to

tendon-derived fibroblasts following treatment with AGEs. There were some doubts about the number of rats in treated (exposed) and control and need clarification. Several varieties were evaluated that showed in 22 figures. Except numerous figures, I didn’t find any fault in the manuscript.

Thank you for your comments. We have added clarification on the source animals used for our study. Cells were isolated from five donor rats. Experiments were completed in a paired fashion such that each donor rat’s tendon fibroblast was exposed to AGE-BSA and BSA-only conditions. 

 

REFERENCES

1. Boehm BO, Schilling S, Rosinger S, Lang GE, Lang GK, Kientsch-Engel R, et al. Elevated serum levels of N(epsilon)-carboxymethyl-lysine, an advanced glycation end product, are associated with proliferative diabetic retinopathy and macular oedema. Diabetologia. 2004;47(8):1376-9.

2. Yoshida N, Okumura K, Aso Y. High serum pentosidine concentrations are associated with increased arterial stiffness and thickness in patients with type 2 diabetes. Metabolism. 2005;54(3):345-50.

3. Patel SH, Yue F, Saw SK, Foguth R, Cannon JR, Shannahan JH, et al. Advanced Glycation End-Products Suppress Mitochondrial Function and Proliferative Capacity of Achilles Tendon-Derived Fibroblasts. Sci Rep. 2019;9(1):12614.

4. Gumucio JP, Schonk MM, Kharaz YA, Comerford E, Mendias CL. Scleraxis is required for the growth of adult tendons in response to mechanical loading. JCI Insight. 2020;5(13).

5. Disser NP, Ghahramani GC, Swanson JB, Wada S, Chao ML, Rodeo SA, et al. Widespread diversity in the transcriptomes of functionally divergent limb tendons. J Physiol. 2020;598(8):1537-50.

6. Liu Y, Zhou J, White KP. RNA-seq differential expression studies: more sequence or more replication? Bioinformatics. 2014;30(3):301-4.

7. Ching T, Huang S, Garmire LX. Power analysis and sample size estimation for RNA-Seq differential expression. RNA. 2014;20(11):1684-96.

---

## [Decision Letter · Decision Letter 1]

7 Jul 2022

Descriptive Transcriptome Analysis of Tendon Derived Fibroblasts Following In-Vitro Exposure to Advanced Glycation End Products

PONE-D-22-07074R1

Dear Dr. Carroll,

We’re pleased to inform you that your manuscript has been judged scientifically suitable for publication and will be formally accepted for publication once it meets all outstanding technical requirements.

Kind regards,

Ming-Chang Chiang

Academic Editor

PLOS ONE

Additional Editor Comments (optional):

Reviewers' comments:

Reviewer's Responses to Questions

**Comments to the Author**

1. If the authors have adequately addressed your comments raised in a previous round of review and you feel that this manuscript is now acceptable for publication, you may indicate that here to bypass the “Comments to the Author” section, enter your conflict of interest statement in the “Confidential to Editor” section, and submit your "Accept" recommendation.

Reviewer #1: All comments have been addressed

Reviewer #3: All comments have been addressed

2. Is the manuscript technically sound, and do the data support the conclusions?

Reviewer #1: Yes

Reviewer #3: Yes

3. Has the statistical analysis been performed appropriately and rigorously? 

Reviewer #1: Yes

Reviewer #3: Yes

4. Have the authors made all data underlying the findings in their manuscript fully available?

Reviewer #1: Yes

Reviewer #3: Yes

5. Is the manuscript presented in an intelligible fashion and written in standard English?

Reviewer #1: Yes

Reviewer #3: Yes

6. Review Comments to the Author

Reviewer #1: I congratulate authors. All comments have been addressed properly. The manuscript is a good shape for publication.

Reviewer #3: Dear author

This study could be one of novel studies that investigate the transcriptome-wide changes in tendon-

derived fibroblasts after treatment with AGEs.The revision was done. Althoguh it could be better if the number of animals clearly noted at animal protocol section that were described five-six wk old rats were selected,than Tendon fibroblast isolation and cell culture section: that number of donor animals were shown in parenthesis (5). However, I think the method were written fluent and clear.

Regards

7. PLOS authors have the option to publish the peer review history of their article (what does this mean?). If published, this will include your full peer review and any attached files.

Reviewer #1: **Yes: **Yuichiro Adachi

Reviewer #3: **Yes: **Akefeh Ahmadiafshar M.D

---

## [Editor Report · Acceptance letter]

15 Jul 2022

PONE-D-22-07074R1 

Descriptive transcriptome analysis of tendon derived fibroblasts following in-vitro exposure to advanced glycation end products 

Dear Dr. Carroll:

I'm pleased to inform you that your manuscript has been deemed suitable for publication in PLOS ONE. Congratulations! Your manuscript is now with our production department. 

Kind regards, 

on behalf of

Dr. Ming-Chang Chiang 

Academic Editor

PLOS ONE